# Step-Level Sparse Autoencoder for Reasoning Process Interpretation

Xuan Yang [* 1 2]   Jiayu Liu [* 2]   Yuhang Lai [* 1 2]   Hao Xu [3]   Zhenya Huang [4]   Ning Miao [1 2]

## Abstract

Large Language Models (LLMs) have achieved strong complex reasoning capabilities through Chain-of-Thought (CoT) reasoning. However, their reasoning patterns remain too complicated to analyze. While Sparse Autoencoders (SAEs) have emerged as a powerful tool for interpretability, existing approaches predominantly operate at the token level, creating a granularity mismatch when capturing more critical step-level information, such as reasoning direction and semantic transitions. In this work, we propose step-level sparse autoencoder (SSAE), which serves as an analytical tool to disentangle different aspects of LLMs' reasoning steps into sparse features. Specifically, by precisely controlling the sparsity of a step feature conditioned on its context, we form an information bottleneck in step reconstruction, which splits incremental information from background information and disentangles it into several sparsely activated dimensions. Experiments on multiple base models and reasoning tasks show the effectiveness of the extracted features. By linear probing, we can easily predict surface-level information, such as generation length and first token distribution, as well as more complicated properties, such as the correctness and logicality of the step. These observations indicate that LLMs should already at least partly know about these properties during generation, which provides the foundation for the self-verification ability of LLMs. Our code is available at https://github.com/Miaow-Lab/SSAE.

*Equal contribution [1]Department of Data Science, City University of Hong Kong [2]Hong Kong Institute of AI for Science, City University of Hong Kong [3]Li Auto Inc [4]State Key Laboratory of Cognitive Intelligence, University of Science and Technology of China. Correspondence to: Ning Miao <ningmiao@cityu.edu.hk>.

*Proceedings of the 43rd International Conference on Machine Learning*, Seoul, South Korea. PMLR 306, 2026. Copyright 2026 by the author(s).

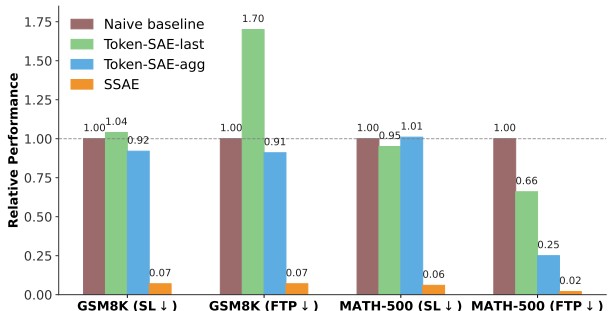

*Figure 1.* We evaluate two step-level tasks: first-token prediction (FTP) and sentence-length (SL) prediction, representing the directional and depth-wise characteristics of a reasoning step. The y-axis reports relative metrics (PPL / RMSE) compared to a statistical baseline (lower is better). Token-based SAEs fail to capture such step-level information, while our SSAE achieves accurate prediction.

## 1. Introduction

Large Language Models (LLMs) have demonstrated remarkable efficacy in complex reasoning tasks, fundamentally shifting the paradigm of problem solving through Chain-of-Thought (CoT) reasoning (Wei et al., 2022; Kojima et al., 2022). By decomposing complex queries into sequences of intermediate deductions, CoT enables models to traverse multi-hop logical paths (Yao et al., 2023). Despite this success, the reasoning mechanism of LLMs remains largely unexplored, because of the complexity of reasoning itself and the diversity of natural language expressions.

Sparse Autoencoders (SAEs) have emerged as a leading technique in interpreting the reasoning process of LLMs (Bricken et al., 2023; Huben et al., 2024; Gao et al., 2025). By projecting dense transformer activations into a high-dimensional and sparse latent space, SAEs facilitate the decomposition of complex neural signals into interpretable, monosemantic features (Shu et al., 2025). While they can peek into the internal working mechanism of transformers, they mainly focus on token-level features. However, step-level features, such as reasoning directions and semantic transitions, are more important to the analysis of LLMs' behaviors. Meanwhile, at each time step, SAE features should encode all information required to successfully reconstruct the activation, which is a mixture of old knowledge that already exists in the previous context and new

information in the current step.

In analyzing LLM behavior, our primary focus lies in the incremental information that emerges at each step, as it directly reflects current-step decisions or preferences. Conversely, pre-existing information from prior steps is often treated as a residual signal that may confound the interpretation of the model's immediate reasoning. However, existing token-based SAEs operate at a token-level granularity, inevitably entangling the incremental knowledge with the contextual background. As shown in Figure 1, when using token-based SAEs to predict step-level information, the perplexity is extremely high, which hinders their applications in interpreting high-level characteristics of LLMs.

To address these problems, we propose SSAE, a framework for interpreting and steering the stepwise reasoning of LLMs. Specifically, SSAE is built upon a context-conditioned Sparse Autoencoder. It is trained to reconstruct each individual reasoning step by conditioning the reconstruction objective on both the global context and a sparse feature vector $\hat{h}$, which extracts the incremental information in the current step. Different from traditional autoencoders, both the encoder and decoder of SSAE have access to contexts, including the query and previous steps. This ensures that $\hat{h}$ encodes only the new information added in the current step. For example, if a number at the current step is copied from the previous step, there is no need to store it again, because we only need to know which number to use. To further squeeze background information out of $\hat{h}$, we restrict its information bandwidth by controlling the sparsity level of $\hat{h}$. In this way, we are able to disentangle information of different reasoning steps into their respective features, further resolving the incremental updates of each step into a set of sparse, monosemantic features $\hat{h}$. Importantly, this representation captures not only the semantics of the current step but also the incremental change it introduces relative to the preceding context—an idea reminiscent of conditional random fields (Lafferty et al., 2001), allowing us to interpret the inherent randomness of generation and distinguish between consistent logical deduction and stochastic hallucination. In decoding, this vector is then translated back into natural language through a step-level decoder.

To quantitatively verify the expressiveness of the learned features $\hat{h}$, we perform probing experiments to directly predict reasoning properties from them. Our results show that SSAE features can accurately predict reasoning characteristics, such as step correctness, logical coherence, and step length. Compared with traditional SAE methods, SSAE can achieve an improvement of accuracy by up to $97.4\%$. Notably, we find that step correctness is also highly predictable from the SSAE features. This suggests that LLMs possess at least partial awareness of the correctness of their reasoning steps prior to generating the actual output. Without care-ful calibration in post-training, LLMs do not know how to leverage this information into their generation.

We further utilize SSAE to uncover latent patterns in LLM reasoning. Our analysis begins by mining frequent activation patterns associated with individual dimensions of $\hat{h}$. We observe that a large proportion of activated dimensions of $\hat{h}$ have a clear pattern to control. Based on this, a deeper analysis can be conducted to check the latent reasoning styles of an LLM. For example, we find that final resolutions are the most frequent patterns in Qwen2.5-0.5B, while reasoning is more frequent in Llama-3.2-1B.

Beyond serving as an analytical tool, SSAE can also be used to boost the reasoning performance of LLMs at inference time. For example, since step correctness can be easily probed from $v$, we can use the predicted correctness as weights in majority voting. Experiments across several reasoning benchmarks and different LLMs show consistent improvements. Moreover, because SSAE is a lightweight model and its encoding process is highly parallelizable, the associated computational overhead is negligible.

The contributions of this work include:

1. We propose SSAE, a framework that interprets LLMs' reasoning dynamics at the step level.

2. Through probing experiments, we demonstrate that SSAE can extract a sparse feature vector $\hat{h}$ that effectively encodes the key reasoning properties.

3. We show that SSAE serves as a versatile framework, providing insights into internal reasoning patterns and enabling enhancements to model performance.

## 2. Related Work

Recent advances in the interpretability of LLM reasoning have sought to elucidate the internal decision-making processes of LLMs. Existing methodologies can be broadly taxonomized into two primary streams: methods that localize specific functional components, and methods that decompose representations into interpretable features. In this work, we focus on analyzing LLM-generated reasoning paths by learning interpretable step-level features.

**Direct Localization and Attribution.** This category encompasses techniques that trace model behaviors back to specific inputs or internal sub-modules. Input attribution methods, such as gradient-based saliency (Achtibat et al., 2024) and perturbation analysis (Cohen-Wang et al., 2024), quantify the marginal contribution of input tokens to the final predictions (Ferrando et al., 2024). Internally, techniques like the Logit Lens (Wendler et al., 2024) project intermediate residual streams into vocabulary space to decode transient

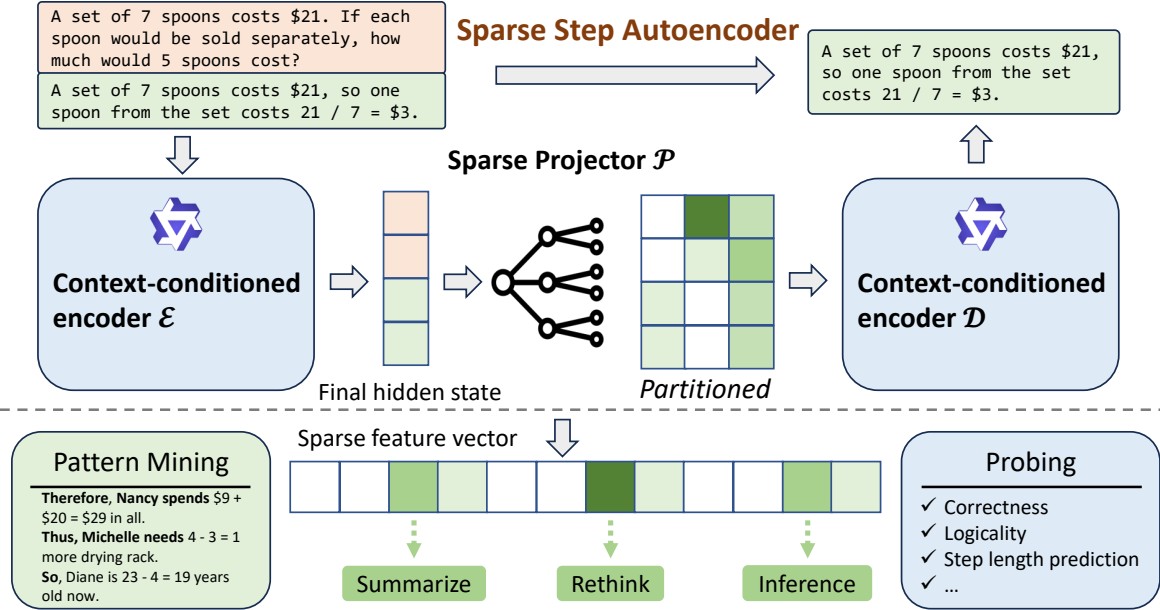

*Figure 2.* Overview of our SSAE framework.

model confidence, revealing which layers or heads influence output probabilities. Beyond correlation, causal intervention methods, including activation patching (Makelov et al., 2024) and causal tracing (Meng et al., 2022), systematically manipulate internal activations to observe downstream effects. While these approaches have successfully identified "circuits" for tasks such as factual recall, they face inherent limitations in semantic interpretability: knowing *where* computation occurs does not necessarily explain *what* concepts are being processed, particularly when operating on dense, polysemantic activation vectors.

**Latent Feature Disentanglement.** To address the inherent opacity of dense representations, Sparse Autoencoders (SAEs) have emerged as a dominant paradigm for dictionary learning within LLMs (Huben et al., 2024; Lan et al., 2025). Predicated on the hypothesis that polysemanticity arises from feature superposition (Rajamanoharan et al., 2024; Liu et al., 2025), SAEs employ sparsity penalties (e.g., $L_1$ regularization or Top-K activation) to decompose dense activations into an overcomplete basis of monosemantic features (Shu et al., 2025; Gao et al., 2025; Bricken et al., 2023). These disentangled representations facilitate precise control, enabling effective model steering by clamping specific feature vectors (Templeton et al., 2024; Turner et al., 2024). However, existing SAE approaches predominantly operate at the token level. This granularity often yields a fragmented view of the reasoning process, failing to capture the high-level propositional logic required for complex deduction. In contrast, our SSAE operates at the step level, encoding the incremental semantic updates essential for interpreting and guiding multi-step reasoning dynamics.

## 3. SSAE: Step-level Sparse Autoencoder

In this section, we introduce SSAE, a step-level sparse autoencoder that extracts interpretable, controllable representations of reasoning steps. We will begin by detailing the architecture and training of SSAE. Subsequently, we introduce the usage of SSAE with a concrete example.

### 3.1. Model Architecture

Traditional SAEs typically contain three components: Encoder, Sparse projector, and Decoder. The encoder maps inputs to a dense representation space, which is subsequently projected into a high-dimensional sparse latent space. Finally, the decoder reconstructs the original input from these sparse features, ensuring representational fidelity.

However, unlike traditional SAEs that deal with independent inputs, SSAE handles sequential and contextually dependent reasoning steps within a reasoning trajectory. When analyzing or manipulating such steps, the key lies in disentangling the incremental information from the background information. From an analytical perspective, the background information is already represented in the feature of previous steps, so there is no need to re-encode it in the current step's representation. From a manipulation perspective, modifying only the incremental information is both necessary and sufficient, because manipulating background information will lead to a mismatch with the context.

To ensure the extracted features $\hat{\mathbf{h}}_k$ contain only incremental information, in SSAE, we enable the decoder to access the full context. In this way, $\hat{\mathbf{h}}_k$ can focus solely on stor-

ing step-specific incremental information, while contextual background information can be easily incorporated during step reconstruction. This context-conditioned formulation ensures that each step is represented in terms of how it advances the reasoning trajectory, rather than as a static semantic unit detached from its history, thereby enabling more precise step-level controllability.

Formally, consider a reasoning trajectory $(s_1, s_2, ..., s_k)$, where $s_k$ denotes the $k$-th step and $C_k = (s_1, ..., s_{k-1})$ denotes its context. SSAE consists of three components: *Context-conditioned encoder* $\mathcal{E}$, *Sparse projector* $\mathcal{P}$, and *Context-conditioned decoder* $\mathcal{D}$. For each step, we concatenate the context and step with a separator token to construct the input of the encoder $x_k = [C_k; |\texttt{SEP}|; s_k]$. The encoder $\mathcal{E}$, instantiated as a Transformer(Vaswani et al., 2017), maps this sequence to a contextualized embedding, where the final hidden state $\mathbf{h}_k$ summarizes the semantic relation between $s_k$ and its preceding trajectory:

$$\mathbf{h}_k = \mathcal{E}([C_k; |\texttt{SEP}|; s_k])_{last} \in \mathbb{R}^d. \tag{1}$$

To disentangle different reasoning factors, we follow sparse autoencoder (Huben et al., 2024; Gao et al., 2025) and pass this vector through the projector $\mathcal{P}$, which expands it into a higher-dimensional vector $\hat{\mathbf{h}}_k \in \mathbb{R}^{cd}$ with the sparsity factor $c \geq 1$. The expansion encourages individual dimensions of $\hat{\mathbf{h}}_k$ to align with atomic, mono-semantic features.

$$\hat{\mathbf{h}}_k = \text{ReLU}(W \cdot \mathbf{h_k} + \mathbf{b}). \tag{2}$$

Decoding is also context-conditioned. Instead of reconstructing $s_k$ from $\hat{\mathbf{h}}_k$ alone, our decoder $\mathcal{D}$ integrates the contextual embeddings of $C_k$ with the latent features $\hat{\mathbf{h}}_k$. Specifically, $\hat{\mathbf{h}}_k$ is partitioned into $c$ segments of dimension $d$ $(\tilde{\mathbf{h}}_k^{(1)}, ..., \tilde{\mathbf{h}}_k^{(c)})$ and appended to the embedding sequence of $C_k$, yielding a combined input of length $|C_k| + c + 1$. The decoder then autoregressively reconstructs the step:

$$\hat{s}_k = \mathcal{D}([Embed(C_k; \texttt{[SEP]}); \tilde{\mathbf{h}}_k^{(1)}, \tilde{\mathbf{h}}_k^{(2)}, ..., \tilde{\mathbf{h}}_k^{(c)}]). \tag{3}$$

### 3.2. Training

The training of SSAE involves optimizing two complementary objectives: (1) Reconstruction Loss $\mathcal{L}_{reconstruct}$ that ensures the extracted feature contains all incremental information to reconstruct the original reasoning step $s_k$, which is implemented as the cross-entropy between the predicted tokens in $\hat{s}_k$ and the ground-truth $s_k$. Let $s_k = (t_1, t_2, ..., t_L)$ be a sequence of $L$ tokens, the loss is formulated as the average negative log-likelihood over the tokens:

$$\mathcal{L}_{reconstruct} = -\frac{1}{L} \sum_{i=1}^{L} \log P(t_i | C_k, \tilde{\mathbf{h}}_k^{(1)}, ..., \tilde{\mathbf{h}}_k^{(c)}, t_{<i}). \tag{4}$$

(2) Sparsity Loss $\mathcal{L}_{sparsity}$ enforces that only a small subset of dimensions in $\hat{\mathbf{h}}_k$ are active, which encourages the representation to be compact and disentangled. In addition, it forms an information bottleneck to avoid redundant background information from being encoded into $\hat{\mathbf{h}}_k$. This is formulated as a standard $l_1$ penalty:

$$\mathcal{L}_{sparsity} = ||\hat{\mathbf{h}}_k||_1. \tag{5}$$

The overall training objective is therefore

$$\mathcal{L} = \mathcal{L}_{reconstruct} + \lambda \cdot \mathcal{L}_{sparsity}, \tag{6}$$

where $\lambda$ is a hyper-parameter that controls the strength of sparsity loss.

To avoid the hassle of tuning $\lambda$, we use a dynamic weight controller to automatically tune $\lambda$ as training proceeds, which is motivated by (Miao et al., 2023). The basic idea is that we first pre-define a target sparsity $\tau_{spar}$: if the actual sparsity is higher than $\tau_{spar}$, $\lambda$ is increased; otherwise, it is decreased. In practice, to achieve smooth control of sparsity in the stochastic training process, we compute the running average of the sparsity, denoted as $\overline{\mathcal{L}}_{sparsity}^{(t)}$, over an update window of size $N$. At the end of the $t$-th window, $\lambda$ is updated multiplicatively based on the deviation of the observed sparsity from $\tau_{spar}$. To prevent numerical instability or collapse during early training phases, the coefficient is constrained within a robust range $[\lambda_{min}, \lambda_{max}]$. The whole update logic is formalized as:

$$\overline{\mathcal{L}}_{sparsity}^{(t)} = \frac{1}{N} \sum_{k=(t-1)N+1}^{tN} \mathcal{L}_{sparsity}(\hat{\mathbf{h}}_k),$$

$$\lambda^{(t+1)} = \text{clip}\Big(\lambda^{(t)} \cdot \big(1 + \alpha \cdot \text{sgn}(\overline{\mathcal{L}}_{sparsity}^{(t)} - \tau_{spar})\big),$$

$$\lambda_{\min}, \lambda_{\max}\Big), \tag{7}$$

where $\alpha$ represents the adjustment step size, which is 0.01 in all our experiments. This feedback loop ensures that the model converges to a latent representation that adheres to the desired information density constraints, independent of the initial scale of the reconstruction loss.

To ensure that the learned features are robust to tiny numerical differences, we also add a Gaussian noise $\epsilon \sim \mathbb{N}(0, \sigma^2)$ to each dimension in $\hat{\mathbf{h}}_k$, where $\sigma = 0.01$. For a sparsity $\tau_{spar}$ and a noise level $\sigma$, maximum information bandwidth can be estimated as

$$\text{IB} = \binom{\frac{\tau_{spar}}{4\sigma}}{cd}, \tag{8}$$

assuming a tolerated feature transmission error of $5\%$, which finally forms the information bottleneck in SSAE.

*Table 1.* Predictive performance of classifiers across various inputs on GSM8K and MATH-500 benchmarks for four reasoning attributes. $\hat{\mathbf{h}}_k$ denotes the high-dimensional sparse latent vector in the feature space, whereas $\mathbf{h}_k$ represents the original dense representation.

| Model | Feature | Correctness Acc ↑ | | Logicality Acc ↑ | Step Length Error ↓ | | First Token Perplexity ↓ | |
|---|---|---|---|---|---|---|---|---|
| | | GSM8K | MATH-500 | MATH-500 | GSM8K | MATH-500 | GSM8K | MATH-500 |
| Naive baseline | - | 70.49 | 70.65 | 55.06 | 28.04 | 33.30 | 61.01 | 74.96 |
| Token-SAE-last | $\hat{\mathbf{h}}_k$ | 72.44 | 86.79 | 60.56 | 29.06 | 31.58 | 103.54 | 49.17 |
| Token-SAE-agg | $\hat{\mathbf{h}}_k$ | 74.38 | **86.88** | 67.43 | 25.79 | 30.33 | 61.55 | 18.63 |
| SSAE-Qwen | $\hat{\mathbf{h}}_k$ | 78.58 | 82.74 | **76.56** | 2.10 | 1.94 | 4.09 | **1.46** |
| | $\mathbf{h}_k$ | 72.30 | 74.52 | 70.35 | 22.71 | 30.35 | 16.75 | 15.91 |
| SSAE-Llama | $\hat{\mathbf{h}}_k$ | **80.55** | 86.24 | 71.91 | **2.02** | **1.59** | **2.66** | 1.62 |
| | $\mathbf{h}_k$ | 73.42 | 79.11 | 63.35 | 20.55 | 29.17 | 19.00 | 15.95 |

## 4. Experiments and Applications

In this section, we conduct a comprehensive empirical evaluation of SSAE to assess its interpretability and downstream utility. First, we quantify the representational fidelity of the sparse features learned by SSAE via a series of diagnostic probing classifiers (Sec.4.2). Subsequently, we characterize the semantic landscape of SSAE features by employing the Neuron-to-Graph (N2G) framework to map latent dimensions to human-understandable reasoning patterns (Sec.4.3). Then, we elucidate the correspondence between distinct reasoning trajectories and the activation of SSAE features. By manipulating specific dimensions, we intuitively validate the controllability of SSAE features in modulating the model's reasoning path and direction.(Sec. 4.5). Finally, leveraging the predictive characteristics of SSAE features, we demonstrate that SSAE can be utilized to augment the reasoning capabilities of LLMs at inference time (Sec.4.6).

### 4.1. Experiment Setup

**Training details.** We train our framework on three datasets: (a) GSM8K-Aug(Deng et al., 2023): A large-scale synthetic expansion of the canonical GSM8K(Cobbe et al., 2021) benchmark. By utilizing GPT-4 to generate semantic perturbations and diverse reasoning paths, this dataset expands the original training set to approximately 385K samples, exposing the model to a wide distribution of linguistic variations essential for robust feature learning. (b) NuminaMath-CoT(Li et al., 2024): A comprehensive reasoning corpus comprising approximately 860K mathematical problems sourced from Chinese K-12 exams and international Olympiad competitions (e.g., AIME, AMC). All solutions are standardized into a unified Chain-of-Thought format, providing the complex, multi-step logical structures necessary to capture advanced deductive dynamics. (c) OpenCodeInstruct(Ahmad et al., 2025): A large-scale, multi-turn instruction-tuning dataset designed to enhance the code generation and reasoning capabilities of large lan-

guage models through execution-based feedback and iterative refinement. Consistent with our step-level framework, we partition the continuous reasoning trajectories of these three datasets into discrete, atomic steps based on natural linguistic boundaries (e.g., punctuation or line breaks) to facilitate independent training of SSAE.

**Implementation Details.** To evaluate our framework across different architectures, we instantiate two distinct variants: **SSAE-Qwen**, where the context-conditioned encoder and decoder are initialized from Qwen2.5-0.5B(Qwen et al., 2025), and **SSAE-Llama**, which utilizes Llama-3.2-1B (Grattafiori et al., 2024) as the backbone. Regarding the sparse feature space, we set the sparse factor $c = 1$ and target sparsity $\tau_{spar} = 10$. To enforce this constraint, the dynamic weight controller continuously modulates the regularization coefficient $\lambda$ during training, ensuring that the empirical sparsity on the validation set converges to and stabilizes at the predefined sparse level $\tau_{spar}$.

### 4.2. Probing with Classifier

To further investigate the expressiveness of the learned sparse feature vectors $\hat{\mathbf{h}}_k$, we hypothesize that several critical meta-reasoning features such as specifically step validity (e.g., correctness, logicality) and the planning horizon (e.g. step length, first token of the step) are not merely emergent properties of the final output, but are explicitly decodable from the intermediate sparse activations.

Formally, for a given latent vector $\hat{\mathbf{h}}_k$ corresponding to step $s_k$, we train a suite of diagnostic classifiers $f_\phi$. The probing objective is to minimize the prediction error against a set of ground-truth step attributes $\mathcal{Y}$:

$$\min_\phi \mathcal{L}_{\text{probe}} \left( f_\phi(\hat{\mathbf{h}}_k), y \right). \tag{9}$$

We adopt a three-layer MLP architecture to probe $\hat{\mathbf{h}}_k$ for four essential meta-reasoning attributes: (1) Step Correctness, which identifies the logical validity of the $\hat{s}_k$; (2)

*Table 2.* Evaluation of N2G pattern extraction fidelity across different sparsity targets $\tau_{spar}$ for SSAE-Qwen.

| $\tau_{spar}$ | GSM8K | | | Numina | | | OpenCodeInstruct | | |
|---|---|---|---|---|---|---|---|---|---|
| | precision | recall | F1 | precision | recall | F1 | precision | recall | F1 |
| 10 | 79.00 | 80.00 | 79.49 | 74.23 | 72.45 | 73.33 | 70.09 | 83.47 | 76.20 |
| 5 | 59.40 | 62.75 | 61.03 | 53.25 | 60.96 | 56.84 | 42.19 | 55.10 | 47.79 |
| 3 | 45.50 | 62.69 | 52.73 | 52.31 | 82.51 | 64.03 | 32.77 | 41.95 | 36.80 |

Logicality, which evaluates the semantic tie between $\hat{s}_k$ and its history $C_k$ regardless of correctness; (3) Step Length: a regression target predicting the token count of $\hat{s}_k$; and (4) First Token Perplexity, which is used to quantify the model's generation confidence.

We evaluate the robustness of SSAE across architectures by probing SSAE-Qwen and SSAE-Llama backbones. To generate training labels, we decode sparse features into the step-level space and extract semantic attributes. For in-domain analysis on GSM8K, we use feature vectors from GSM8K-Aug-trained models, with ground-truth labels derived programmatically via symbolic verification of numerical outputs. For out-of-distribution (OOD) context, we employ representations from models trained on NuminaMath-CoT and assess performance on the MATH-500 benchmark. Due to the open-ended and proof-centric nature of these problems, which often lack explicit symbolic annotations, we employ an LLM-as-a-judge. Specifically, we leverage GPT-4o-mini to generate binary assessments of step correctness and logicality relative to the preceding context.

We compare the performance of SSAE with traditional Token-SAE and naive statistical baselines. Since Token-SAE extracts sparse latents at the token level, there is an inherent granularity mismatch for predicting step-level attributes. To construct a fair comparison, we derive two step-level adaptations. Specifically, Token-SAE-last directly utilizes the sparse feature of the last token in each reasoning step. Recognizing that the information contained within a single token may be insufficient to predict the comprehensive information of the entire reasoning step, we additionally formulate Token-SAE-agg, which employs a Transformer to aggregate the token representations across the entire step. Subsequently, both the last-token sparse vector and the aggregated sparse vector are probed with the same three-layer MLP to predict step-level attributes. For the binary classification tasks, namely Step Correctness and Logical Coherence, we adopt a majority-class baseline determined by the marginal label prevalence within the dataset. For the Step Length prediction, the naive baseline is to directly use the average length of reasoning steps in the dataset as a constant length prediction. Finally, for the First Token Perplexity, we derived the empirical distribution of the first tokens from the training set and subsequently computed

the cross-entropy based on this distribution to establish a stochastic baseline, representing the theoretical performance of a context-agnostic model.

**Results and Analysis.** As shown in Table 1, Token-SAE features cannot be used to predict most of the step-level information, with similar performance as naive statistical baselines. In contrast, SSAE features contain rich step-level information, which can be decodable by a linear probe. For example, SSAE features can almost perfectly predict the step length and first token, which indicates that randomness in expression has been well-modeled by SSAE. SSAE features can also be used to predict step correctness and logicality, which are higher-level and more comprehensive attributes of reasoning steps, achieving at least 10% accuracy increase compared with naive majority-class prediction. SSAE's ability to predict step correctness can be further used to verify LLM-generated reasoning, and, in turn, improve reasoning performance, which we will talk about in Sec.4.6.

Besides comparison with Token-SAE, we also compare SSAE features $\hat{\mathbf{h}}_k$ with dense features $\mathbf{h_k}$ before dimensionality elevation. We find that sparse features $\hat{\mathbf{h}}_k$ are more effective in all downstream tasks, which demonstrates the benefit of information disentanglement and the disposition of interfering background information.

### 4.3. N2G Pattern Mining

To translate the extracted sparse latent features into human-interpretable concepts, we use the Neuron to Graph (N2G) framework (Foote et al., 2023) as a tool to mine frequent patterns related to each dimension of SSAE features.

To compare the characteristics of features learned under different sparsity constraints $\tau_{spar}$, we first apply N2G to analyze three distinct SSAE-Qwen variants with $\tau_{spar} \in \{3, 5, 10\}$. Specifically, we measure the alignment between the graph-predicted activations based on linguistic patterns and the ground-truth latent firings via precision, recall, and F1 score. Higher F1 scores indicate more biunique mapping between patterns and feature dimensions. We find that SSAE with $\tau_{spar} = 10$ consistently achieves the highest F1 scores across all benchmarks, indicating that a higher dimensionality is required to encode the nuance of reasoning steps into monosemantic units. On the contrary, when

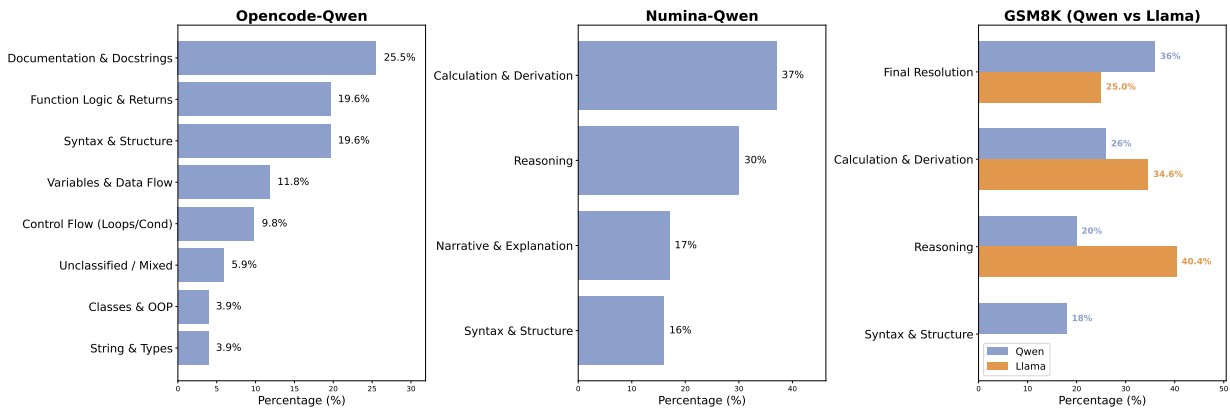

*Figure 3.* Taxonomy and statistical distribution of N2G patterns across various SSAE configurations and datasets.

$\tau_{spar} = 3$, SSAE is forced to compress multiple distinct reasoning concepts into a single latent dimension, causing the N2G graph to squeeze irrelevant patterns into the same dimension, which leads to a drop in F1 scores.

We employed Gemini-3-Pro to systematically analyze and synthesize patterns across various dimensions, subsequently assigning a descriptive label comprising three to six words to each. These labels were then clustered, with semantically similar terms merged to consolidate the findings. The resulting statistical distribution is presented in Figure 3. As illustrated, the semantic distribution of learned features is highly domain-dependent. In the code domain, the model prioritizes structural integrity, with Documentation and Syntax jointly constituting over 45% of the observed features. Conversely, the analysis of mathematical datasets shift its focus toward execution mechanics.

Most notably, the comparative analysis on GSM8K reveals a divergence in latent attention between model families. While Llama-3.2-1B exhibits a dominant emphasis on Reasoning (40.4%), Qwen2.5-0.5 maintains a more uniform distribution across reasoning (20%), calculation, and resolution. This suggests that while Llama's representations on this task are attuned to the explicit chain of thought, Qwen adopts a more balanced approach, weighing the structural and mechanical components of the solution equally with the logical narrative.

*Table 3.* Statistics of SSAE features on GSM8K.

| model | Activation Ratio | Jaccard Similarity |
|---|---|---|
| SSAE-Qwen | 29.01% | 0.6716 |
| SSAE-Llama | 15.65% | 0.3052 |

### 4.4. Disentanglement of Incremental Information

To verify whether SSAE can extracts the incremental information of the current reasoning step while squeezing back-

ground information, we conduct a latent-swap test. Specifically, we first randomly sample a reasoning step and encode it under its original context $C_k$ to obtain the corresponding sparse feature vector $\hat{h}$. Subsequently, we decode $\hat{h}$ under a distinct context $C_j$ $(j \neq k)$. If the sparse feature vector truly captures only the incremental information of the source step, the decoded text should not repeat or paraphrase the source background. Conversely, if background information remains entangled within the latent representation, it will leak into the decoded text.

We employ an LLM-as-a-judge framework to evaluate the semantic overlap between the decoded text and the source background. As shown in Table 4, the decoded content exhibits a high degree of independence from the source background across all evaluated datasets. Furthermore, the evaluations across multiple LLM judges demonstrate exceptional consistency, yielding an observed agreement between 94.87% and 98.75%, and a Gwet's AC1 coefficient between 0.96 and 0.99. This robustly demonstrates that $\hat{h}$ exclusively preserves incremental information, demonstrating the superior disentanglement capability of SSAE. The detailed prompt for LLM-as-a-judge can be found as Appendix B.

### 4.5. Feature Statistics and Manipulation

Table 3 shows some key statistics of SSAE features of a diverse group of reasoning paths. For each feature, $15.65\%$ to $29.01\%$ dimensions are activated on average, showing the strong sparsity of learned features. The Jaccard Similarity between features indicates that some dimensions are shared between, and others are unique to certain steps.

To interpret the distinct roles of both types of dimensions, we conducted a qualitative analysis on representative cases. As shown in Case A (left panel of Figure 4), perturbing the shared dimensions reveals that they predominantly encode more general surface-level stylistic attributes, such as lexical choice and syntactic structure, which remain consistent

| Dataset | Model | GPT-5 | Gemini-2.5-pro | Deepseek-R1 | Observed Agreement | Gwet's AC1 |
|---------|-------|-------|----------------|-------------|--------------------|--------------|
| GSM8K | SSAE-Qwen | 4.96 | 4.95 | 4.91 | 94.87% | 0.96 |
| | SSAE-Llama | 4.99 | 4.99 | 4.98 | 98.11% | 0.98 |
| NuminaMath-CoT | SSAE-Qwen | 4.92 | 4.94 | 4.94 | 97.86% | 0.98 |
| | SSAE-Llama | 4.95 | 4.97 | 4.97 | 98.75% | 0.99 |

*Table 4.* Quantitative results of the latent-swap test. Scores (1–5) evaluate semantic independence, where 5 means fully independent and 1 means fully redundant.

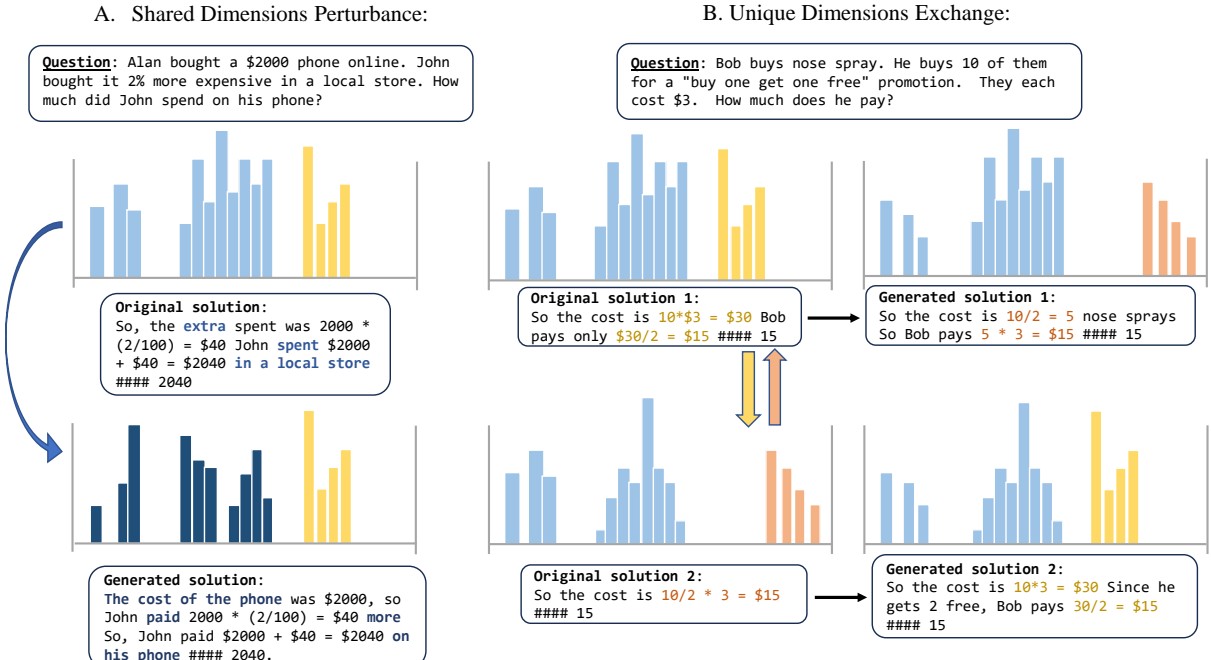

*Figure 4.* Case studies on SSAE feature perturbations. Modulating shared dimensions induces surface-level linguistic variations, whereas the exchange of unique dimensions triggers a crossover of underlying reasoning strategies.

across different reasoning paths. Conversely, exchanging the unique dimensions in Case B (right panel of Figure 4) directly alters deep, reasoning-specific attributes, such as the order of operations or the underlying reasoning direction. This disentanglement further suggests that SSAE can effectively distinguish between the phrasing of a step and its underlying logical content.

### 4.6. Probing Guided Weighted Voting

Extending the utility of our framework, we leverage the predictive fidelity of sparse feature vectors to refine the model's inference capabilities. Since $\hat{\mathbf{h}}_k$ provides a reliable proxy for the validity of reasoning steps, we employ the predicted correctness scores to upgrade the Self Consistency(Wang et al., 2023) baseline from a naive, unweighted majority voting to a quality-weighted inference strategy(Liu et al., 2026). In our approach, we utilize the trained correctness probe $P_\phi(\text{correct}|\hat{\mathbf{h}}_k)$ to assign a confidence weight to each generated step. Let $\mathcal{A}$ be the set of answers derived from $M$ sampled reasoning steps $s_{k+1}^{(1)}, ..., s_{k+1}^{(M)}$ under $C_k$, the

score for a candidate answer $a$ is computed as the sum of the quality-weighted votes:

$$\text{Score}(a) = \sum_{i=1}^{M} \mathbb{I}(s_{k+1}^{(i)} = a) \cdot \left( p_{\text{correct}}^{(i)} \right)^\tau, \quad (10)$$

where $p_{\text{correct}}^{(i)}$ is the predicted correctness probability of $s_{k+1}^{(i)}$ for $f_\phi$, and $\tau \geq 0$ is a temperature hyperparameter balancing the effect of multiple steps. When $\tau = 0$, the method degenerates to self-consistency. As $\tau$ increases, the inference engine progressively down-weights paths with low semantic validity, effectively filtering out reasoning steps that the probe identifies as hallucinations.

**Evaluation Benchmarks.** We assess the effectiveness of Probe-Guided on six comprehensive benchmarks: (1) GSM8K(Cobbe et al., 2021): The canonical benchmark for multi-step grade-school mathematics, serving as the primary in-domain evaluation for our probes. (2) SVAMP(Patel et al., 2021): A benchmark containing 4138 arithmetic problems constructed by applying controlled, semantically meaning-

ful variations to existing single-step arithmetic word problems. (3) MultiArith(Roy & Roth, 2015): A collection of 600 problems for multi-step arithmetic math word problems designed to evaluate a model's ability to perform compositional numerical reasoning over natural language.(4) MATH-500(Hendrycks et al., 2021): A curated subset of the MATH benchmark containing 500 high-school level problems across algebra, geometry, and calculus. (5) AIME-24/25(Art of Problem Solving, 2025): Problems from the American Invitational Mathematics Examination.

**Baselines.** We benchmark our method against two standard decoding baselines: Average Single-Path Accuracy (Avg@16) over 16 samples, representing the expected performance of an individual trajectory, and Self-Consistency (SC@16), which serves as a representative baseline for inference-time scaling via majority voting. To ensure a commensurate computational budget, we fix the number of sampled paths to $k = 16$ across all evaluated methods.

**Evaluation Procedure.** We evaluate the method in two different settings. In the first setting, we use Qwen2.5-0.5B and Llama-3.2-1B, which SSAE is trained from, as base models, to investigate the effectiveness of SSAE in the self-enhancement of reasoning ability. Limited by the capability of the base models, we run our evaluations on 3 relatively easy tasks, including GSM8K, SVAMP, and MultiArith. To further test the cross-model transferability of SSAE, we employ the classifiers based on SSAE-Qwen backbone to score outputs from larger, frontier-grade models, including Qwen2.5-7B-Instruct(Qwen et al., 2025) and DeepSeek-R1-Distill-Qwen-32B(Guo et al., 2025) on more challenging MATH-500 and AIME 2024/2025 benchmarks. This setting provides a critical testbed of whether sparse reasoning features learned from smaller models can generalize to effectively verify the reasoning from more powerful models.

*Table 5.* The evaluation of Probe-Guided on Qwen2.5-0.5B and Llama-3.2-1B.

| Model | Strategy | GSM8K | SVAMP | MultiArith |
|---|---|---|---|---|
| Qwen2.5-0.5B | Avg@16 | 30.40 | 19.33 | 35.56 |
| | SC@16 | 46.20 | 29.67 | 59.44 |
| | PG@16 (ours) | **46.80** | **33.00** | **61.67** |
| Llama-3.2-1B | Avg@16 | 12.00 | 32.33 | 67.22 |
| | SC@16 | 16.60 | 48.00 | 82.78 |
| | PG@16 (ours) | **19.40** | **50.33** | **83.89** |

**Results and analysis.** Table 5 presents the results of applying our Probe-Guided (PG) decoding strategy to SSAE. Across both SSAE-Qwen and SSAE-Llama backbones, PG strategy consistently outperforms the Self-Consistency baseline on all evaluated benchmarks. These results further demonstrate that the sparse feature vector inherently encapsulates the essential reasoning characteristics of each step, and that such attributes are highly predictable through simple linear probes. Empirical results on larger models are demonstrated in Appendix A.

## 5. Conclusion

In this work, we propose a step-level sparse autoencoder (SSAE) to address the difficulty of analyzing LLMs' complex reasoning patterns, resolving the granularity mismatch of existing SAE-based interpretability methods. By conditioning step feature sparsity on context and building an information bottleneck, SSAE effectively disentangles incremental reasoning information from background noise into sparse, interpretable features. Experiments across multiple base models and reasoning tasks validate these features, as linear probing can accurately predict both surface-level attributes and complex reasoning properties (e.g., correctness, logicality). These findings confirm that LLMs encode such properties during reasoning, laying a foundation for their self-verification capabilities and advancing the interpretability of LLM reasoning.

## Impact Statement

This paper presents work whose goal is to advance the field of machine learning. There are many potential societal consequences of our work, none of which we feel must be specifically highlighted here.

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

## A. Performance of Probe-Guided Weighted Voting on Large Model

Table 6 highlights the promising potential of the Probe-Guided strategy within the cross-model supervision framework. Notably, leveraging the SSAE to guide the significantly larger DeepSeek-R1-Distill-Qwen-32B yields a substantial performance gain on AIME 2024, improving accuracy from 86.67% to 90.00%. A marginal enhancement is also observed when guiding Qwen2.5-7B on the MATH-500 benchmark. However, this strategy exhibits clear performance plateaus on exceptionally challenging tasks (e.g., AIME 2025) or when the target model approaches performance saturation (e.g., DeepSeek's 95.60% on MATH-500), where it fails to surpass the Self-Consistency (SC) baseline. We attribute these limitations to representational capacity mismatch and out-of-distribution (OOD) generalization gaps.

*Table 6.* The evaluation of Probe-Guided on larger, frontier-grade models.

| Model | Strategy | MATH-500 | AIME 2024 | AIME 2025 |
|---|---|---|---|---|
| Qwen2.5-7B-Instruct | Avg@16 | 74.00 | 12.08 | 6.46 |
| | SC@16 | 80.20 | 16.67 | 13.33 |
| | PG@16 (Ours) | **80.80** | 16.67 | 13.33 |
| Deepseek-R1-Distill-Qwen32B | Avg@16 | 94.30 | 72.60 | 42.92 |
| | SC@16 | 95.60 | 86.67 | 66.67 |
| | PG@16 (Ours) | 95.60 | **90.00** | 66.67 |

## B. Prompt Template for LLM-as-a-Judge in Latent-Swap Test

Figure 5 proposes the detailed prompt template we used for LLM-as-a-Judge in Latent-Swap Test. The judging prompt explicitly instructs the evaluator to focus on repeated facts, logic, and numerical claims rather than superficial word overlap.

---

**Latent-Swap Test**

```
You are an expert evaluator.  Your task is to assess whether a generated text
(rep_contain) repeats or paraphrases the background information provided in
hint_rep.
## Background
In our experiment, a latent representation is extracted from a source reasoning
step whose preceding context is hint_rep.  This latent is then decoded into text
(rep_contain) under a different problem context.  Ideally, the latent should encode
only the incremental reasoning content of the source step, not the redundant
background information from hint_rep.  If rep_contain is largely independent of
hint_rep, it suggests the latent has successfully captured only the new reasoning
and stripped away the source background.  Conversely, if rep_contain reproduces or
paraphrases content from hint_rep, it means the latent has leaked source-context
information and failed to disentangle reasoning from background.
## Scoring Criteria (1-5 scale)
- 5 (Fully Independent):  rep_contain introduces entirely new information,
calculations, or reasoning steps that have NO semantic overlap with hint_rep.  The
generated text is completely distinct from the source background.
- 4 (Mostly Independent):  rep_contain is largely novel, with only trivial or
unavoidable overlap (e.g., reusing a common number or entity that happens to appear
in hint_rep).
- 3 (Partially Overlapping):  rep_contain contains a mix of new content and some
rephrased or repeated content from hint_rep.
- 2 (Mostly Redundant):  rep_contain largely repeats, paraphrases, or restates the
information already present in `hint_rep`, with only minor new additions.
- 1 (Fully Redundant):  rep_contain is essentially a copy or close paraphrase of
hint_rep, adding no new information.
## Important Notes
- Focus on semantic overlap, not surface-level word matching.  Even if different
words are used, restating the same fact or logic counts as repetition.
- Numbers or entities that naturally must appear in both (e.g., a common quantity)
should NOT be penalized.  Only penalize when rep_contain restates the reasoning,
explanation, or descriptive content from hint_rep.
- Focus on logic and factual statements.  Repetition of trivial function words,
connectives, or common phrasing (e.g., "so", "therefore", "we get") does NOT count
as redundant information.  Only repeated substantive facts, logical steps, or
numerical claims should be considered overlap.
- If rep_contain is incoherent or nonsensical, but does not repeat hint_rep, it
should still receive a high score (since the goal is to measure independence from
the source background, not generation quality).
## Input Format
hint_rep:  {hint_rep}
rep_contain:  {rep_contain}
## Output Format
Respond in the following JSON format only:
```json
"reasoning":  "<brief explanation of your assessment>",
"score":  <integer from 1 to 5>
```
```

*Figure 5.* Prompt Template for llm-as-a-judge in latent-swap test

