# OpenReview forum: "Step-Level Sparse Autoencoder for Reasoning Process Interpretation"
_ICML.cc/2026/Conference — ICML 2026 regular_

### Official Review · Reviewer_goJi · 2026-02-28

**Soundness:** 3
**Presentation:** 3
**Significance:** 1
**Originality:** 3
**Overall Recommendation:** 5
**Confidence:** 4

**Summary:**

Targeting on step-dependent reasoning process, this paper propose SSAE to replace traditional token-level Sparse AutoEncoders. The improved decoder can receive context embeddings to distinguish incremental information from the background information. Experiments indicate that SSAE can provide useful interpretation and help enhance reasoning capabilities.

**Compliance With Llm Reviewing Policy:**

Affirmed.

**Final Justification:**

My final recommendation is to accept this paper considering its quality.

In the initial version, the paper has obtained good improvement to support the significance, and the originality was also good mostly. So I raise some questions regarding the soundness and representation.

In the rebuttal phase, more experiments are introduced to further enhance the soundness, as demonstrated in the rebuttal record. Also, my concerns about the representation (W1) are carefully addressed by adding more detailed explanation.

**Key Questions For Authors:**

See the weaknesses part.

**Limitations:**

Yes

**Strengths And Weaknesses:**

**Strengths**
1.	The proposed SSAE can yield useful features, which is examined through correlation and causality test.
2.	The obtained features are shown to be effective to enhance the reasoning process in both math and code scenarios.
3.	The motivation of step-level SAE aligns well with the step-dependent reasoning scenarios. I’d like to see more tools for interpret large reasoning models.
**Weaknesses**
1.	The detailed architecture of decoder $\mathcal{D}$ should be clarified more precisely. Particularly, what is the shape of $\hat{s}_k$ and is the decoder a linear model or something else? How is Embed(C_k; [SEP]) calculated and what is its shape? The same problem happens when introducing how SSAE helps enhance reasoning ability. Without this information, the paper may not be easy to follow and reproduce, and that no complementary description or implementation is provided further strengthen my concern.
2.	Can authors provide N2G evaluation result on traditional token-level SAE? There seems no baseline to compare with.
3.	The authors claim that SSAE aims to disentangle the incremental information from the background information (in Line 135), which is not verified through empirical experiments. Some visualization results should be presented to support.

I would raise my score once my concerns are addressed :)

---

> ### Author Rebuttal · Authors · 2026-03-31
>
> We extend our sincere gratitude for your meticulous and constructive feedback. Your insightful observations and valuable recommendations have greatly contributed to improving the rigor and clarity of our work!
>
> > **W1: Details of architecture and calculation.**
>
> Thank you for your question!
> 1. Architecture of Decoder $D$:
>
> The role of the $D$ is to reconstruct the current reasoning step $s_k$ conditioned on both the preceding context $C_k$ and the sparse latent feature $\hat{h}_k$. In other words, $D$ takes the contextual embeddings of the previous steps together with the latent of the current step, and outputs the reconstruction token sequence. Therefore, $D$ is **not** a linear model; rather, it shares the same **Transformer** architecture as the Encoder $E$. As mentioned in Section 4.1, both the $E$ and $D$ are instantiated from pretrained LLM base models (Qwen2.5-0.5B and Llama3.2-1B).
>
> 2. Calculation and Shape of $Embed(C_k,[SEP])$:
>
> The term $Embed$ represents the embedding of the last layer hidden states in the Transformer. Specifically, it is the last layer hidden state of $E$ when fed the prefix token sequence $(C_k, [SEP])$. The shape is (batch_size, $|C_k|$ + 1, hidden_size), where $|C_k|$ is the token length of the context, the hidden_size corresponds to the hidden size of the initialized LLM, which is 896 for Qwen2.5-0.5B and 2048 for Llama3.2-1B.
>
> 3. Generation and Shape of $\hat{s}_k$:
>
> $\hat{s}_k$ is the reconstructed text sequence of the current reasoning step. It is the final generation output of $D$. As formulated in Equation (3), the Decoder takes the concatenation of $Embed(C_k, [SEP])$ and the partitioned latent sparse features $\tilde{h}_k^{(i)}$, and autoregressively decodes the step tokens until it outputs the `<eos>` token. The shape is (batch_size, max_new_tokens), where max_new_tokens is the dynamically generated length of the reasoning step.
>
> 4. How this applies to enhancing reasoning ability:
>
> During Probe-Guided Weighted Voting, we do not need to run the Decoder to reconstruct $\hat{s}_k$. The workflow is:
> - The base model generates a candidate step $s_k$
> - $(C_k; [SEP]; s_k)$ is passed through the Encoder $E$ and projector $P$ to extract the sparse latent feature $\hat{h}_k$.
> - $\hat{h}_k$ is fed into the pre-trained, lightweight 3-layer MLP probe to predict the correctness score of the step.
> - This score is then used for weighted voting (Equation 10).
>
> Because we bypass the autoregressive generation of the Decoder during inference, the computational overhead introduced by the SSAE for step verification is minimal and highly efficient.
>
> > **W2: N2G results on Token-SAE.**
>
> Thanks for your suggestion! We conducted the N2G evaluation on the Token-SAE baseline using the same evaluation pipeline. The results are illustrated below, G/N denote GSM8K/NuminaMath-CoT, and P/R/F1 denote Precision/Recall/F1:
> | Model | G-P | G-R | G-F1 | N-P | N-R | N-F1 |
> |-|-:|-:|-:|-:|-:|-:|
> | Token-SAE | 71.63 | 37.18 | 48.90 | 72.90 | 55.19 | 62.80 |
> | SSAE-Qwen ($\tau_{spar}=10$) | **79.00** | **80.00** | **79.49** | **74.23** | **72.45** | **73.33** |
>
> In our experiments, Token-SAE is trained on the same backbone (Qwen2.5-0.5B) and datasets as SSAE-Qwen. Because it produces one sparse latent per token, its granularity does not naturally align with step-level analysis. For a fair comparison, we use the sparse feature of the last token in each reasoning step and apply the same N2G pipeline to both models. These results show that SSAE consistently outperforms Token-SAE, which suggests that Token-SAE features are less consistently associated with stable, recoverable linguistic patterns, whereas our SSAE features are easier to summarize into compact pattern graphs.
>
> > **W3: Visualization of incremental information disentanglement.**
>
> Thank you for the helpful suggestion! We have added a case study to visualize the claim. Please kindly refer to the figure at: https://anonymous.4open.science/api/repo/SSAE_Rebuttal/file/disentanglement_case_study.pdf?v=55e22ca3
>
> In the example, Step 1 introduces a new concept (The forks), while Step 2 performs a calculation from available information. We then extract the sparse latent feature of Step 1 and Step 2 separately, and interpret them as encoding two different kinds of incremental information relative to their contexts:
> - the latent of Step 1 corresponds to introducing a new concept.
> - the latent of Step 2 corresponds to performing calculations over already available concepts.
>
> To verify this, we decode with different context–latent pairings. When we use Background + the Step-2 latent, the decoded next step is again a calculation step, and it does **not** introduce new fork-related information (since it is redundant background information for Step 2). In contrast, when we use Background + Step 1 + the Step-1 latent, the decoded next step again behaves like concept introduction rather than another calculation. We will add this case study to the revised paper.

---

> > ### Author Rebuttal · Reviewer_goJi · 2026-04-01
> >
> > I have received point to point response from the author. Most of my concerns, including W2&W3, have been largely addressed by the new experimental results. Thus, the representation (W1) and soundness (W2) of the paper have been improved. Thus, I’ve raised my score to support weak accept.
> >
> > The reason why I don't raise my score higher is that, the response for W3 is not sufficient to me. Specifically, the author's clarification is limited to case study rather than overall assessment. My expectation may be some embedding comparison for the incremental information from the background information, or llm-as-a-judge verification.
> >
> > ---
> > ## Round 2 Response
> >
> > I appreciate the effort the authors have made. The added LLM-as-a-judge results look positive to support the claim that the proposed method disentangles the incremental information from the background information. I'm happy to see the improvement of this paper. Now I'm leaking to raise my score to further support accepting this paper.

---

> > > ### Author Response · Authors · 2026-04-02
> > >
> > > Thank you very much for your constructive feedback and your recognition that most of your concerns have been largely addressed. Following your suggestion, we added an LLM-as-a-judge verification to directly support the disentanglement claim.
> > >
> > > The key idea of the new experiment is a **latent-swap test**. We randomly take a reasoning step from one example, extract its SSAE sparse latent, and then decode it under a **different background context**. If the sparse latent truly captures only the incremental information of the source step, then the decoded text should not repeat or paraphrase the source background. Conversely, if background information is still entangled in the latent, it would leak into the decoded text.
> > >
> > > To quantify this, we ask multiple LLM judges to score the **semantic overlap** between the decoded text and the source background on a 1–5 scale, where **5 means fully independent** and **1 means fully redundant**. The judging prompt explicitly instructs the evaluator to focus on repeated facts, logic, and numerical claims rather than superficial word overlap. The results below clearly demonstrate a strong disentanglement effect, consistent across different datasets, backbones, and even when evaluated with different LLMs. Our prompt is available at: https://anonymous.4open.science/r/SSAE_Rebuttal/prompt.txt
> > >
> > > | Dataset | Model | GPT-5 | Gemini-2.5-pro | Deepseek-R1 | Observed Agreement | Gwet’s AC1 |
> > > |---|---|---:|---:|---:|---:|---:|
> > > | GSM8K | SSAE-Qwen | 4.96 | 4.95 | 4.91 | 94.87% | 0.96 |
> > > | GSM8K | SSAE-Llama | 4.99 | 4.99 | 4.98 | 98.11% | 0.98 |
> > > | NuminaMath-CoT | SSAE-Qwen | 4.92 | 4.94 | 4.94 | 97.86% | 0.98 |
> > > | NuminaMath-CoT | SSAE-Llama | 4.95 | 4.97 | 4.97 | 98.75% | 0.99 |
> > >
> > > Moreover, the judges are highly consistent, with observed agreement between 94.87% and 98.75% and Gwet’s AC1 between 0.96 and 0.99.
> > >
> > > We sincerely hope this addresses your concern. Unlike the qualitative case study, this experiment tests disentanglement at scale and in a stricter way: it directly checks whether background information leaks through the sparse latent during decoding. The near-ceiling independence scores and strong cross-judge agreement support our claim that SSAE’s sparse features largely preserve incremental reasoning content while stripping away background information. We will add this experiment and all our discussions to the revised paper.
> > >
> > > ---
> > >
> > > ## Round 2 Reply
> > >
> > > It was your highly constructive feedback that motivated us to significantly strengthen both our writing and our empirical evaluation. Thank you for your rigorous review and for helping us make this a much stronger paper!

---

### Official Review · Reviewer_SZjo · 2026-03-02

**Soundness:** 4
**Presentation:** 3
**Significance:** 3
**Originality:** 3
**Overall Recommendation:** 5
**Confidence:** 4

**Summary:**

This work presents a sparse autoencoder variant designed to encode steps rather than individual tokens, aiming to build a higher-level semantic embedding of reasoning processes based on the intuition that reasoning is done at the step level rather than the token level.

**Compliance With Llm Reviewing Policy:**

Affirmed.

**Final Justification:**

While I still have some lingering concerns regarding baseline breadth in the main experiment, the empirical results are for the most part convincing and the methodology aims to do something which seems quite a bit more reasonable and useful than token-level encodings.

**Key Questions For Authors:**

1. Figure 1: I assume you are reporting PPL for token prediction and RMSE (SL???) on the step-length prediction?

2. Figure 1: Should first-token perplexity really be low? If you are learning truly semantic encodings of steps, the exact verbalization of that semantic should be flexible: there should be several similarly likely synonymous step verbalizations according to a predictor model which takes the SAE encoding as input. The same can be said for sentence length (although I would concede that the synonymous sentences are likely to be correlated in length based on the complexity of the underlying semantic to be expressed).

3. You give \tau_spar, but don’t explain how it is measured. What does \tau_spar = 10 mean?

4. You need to add some discussion of your baselines to the experimental setting section. For example, which LLM do you use for Token-SAE? You should also include some reasoning as to why you selected your baselines. There are many SAE and logit probing papers, why do you include none but the vanilla token SAE as baselines?

5. Section 4.2: Are you predicting perplexity, or are you predicting the first token and measuring the prediction’s perplexity? At line 256, you say “we adopt a 3-layer MLP architecture to [predict] […] First Token Perplexity”,  which implies that you are actually predicting perplexity and reporting some regression metric on your PPL prediction. Is this the case?

6: I think your naive statistical baseline is too naive. If I understand correctly, your naive binary classifier just picks the most common of the two labels for each prediction? I think a linear classifier over the hidden state of the final layer of the LLM, for example, would be a considerably more appropriate naive baseline than this. The idea behind SAEs in general is that the sparsity makes them better suited to produce meaningful features. Accordingly, a (simple) non-sparse architecture would be considered naive to this rationale. Plus, this would demonstrate your method’s ability to improve on methods which simply use the LLMs information-dense hidden states as text encodings.

7: You should really have a basic description of N2G in your work. I shouldn’t have to read another paper to get a fundamental understanding of your experimental methodology.

8: “We measure the alignment between the graph-predicted activations based on linguistic patters and the ground-truth latent firings”. How do you collect ground truth latent firings? With what do you expect the activations to be aligned with?

9: Table 4:  “These results further demonstrate that the sparse feature vector inherently encapsulates the essential reasoning characteristics of each step”. I don’t think your results, which show marginal improvements on self consistency, demonstrate this claim. If swapping out a different step-scoring method were swapped in for yours in the quality-weighting mechanism you present were to perform even better, would that mean that this other scoring method encapsulates the essential reasoning characteristics to a greater degree?

10: Considering how naive your baseline is in Table 1, it is shocking that the token-SAE baseline is actually at times similar, or worse. Can you explain what is happening in these cases? Would other, more sophisticated probing/encoding methods also perform so naively?

In summary, I think this paper is of acceptable novelty and contribution. However, I am somewhat unconvinced by their experimental results. Each experiment is designed to test a hypothesis, and so the positive outcome of the experiment must be obvious or clearly explained. In Table 1, for example, it is clearly implied: if the proposed method has better performance than the other tested methods then the hypothesis that step-level encoding more aptly captures reasoning and that the proposed methodology can achieve step-level encoding is validated. However, I think that the chosen baselines are so naive that this experiment can not truly validate this hypothesis. Similarly, I am unconvinced that first token and sentence length are meaningful tasks. In Table 3, which is used to demonstrate true sparsity in the learned representations, no control is tested; typically we would expect to see similar measurements on features which are known to not be sparse. If the proposed features show significantly lower activation ratio and similarity, then we can say that they are sparse. If you can improve the credibility of the claims you aim to demonstrate empirically, I will raise my score.

**Limitations:**

Yes

**Strengths And Weaknesses:**

Strengths:
1. Conceptually, the work is well motivated. Indeed, it seems self-evident that reasoning is not done at the token level but in the form of logical (or mathematical) steps involving a complete semantic expression
2. The authors made a clear effort to demonstrate the value of their method via a variety of empirical analyses

Weaknesses:
1. Explanations are often unclear or lacking throughout the paper
2. Despite the variety of experimentation, I find several of the experiments to be insufficiently comprehensive or meaningful to be satisfactory

---

> ### Author Rebuttal · Authors · 2026-03-31
>
> We sincerely thank you for your thoughtful and constructive feedback! As for your concerns:
> > **Q1: Figure 1 notations.**
>
> Yes, Figure 1 reports PPL for first-token prediction and RMSE for sentence-length (SL) prediction. We will clarify the x-axis labeling in the revision.
>
> >**Q2: Why first-token perplexity should be low?**
>
> Thanks for the question! Although a reasoning step may admit multiple valid verbalizations, its first token is usually less diverse because it often signals the reasoning direction (e.g., continuation or conclusion). Moreover, the perplexity metric measures the reconstruction of the observed step, not the total probability over semantically equivalent verbalizations.  Under this interpretation, lower first-token perplexity indicates that the latent feature preserves information that makes the observed step easier to reconstruct.
>
> >**Q3: Explanation of $\tau_{spar}$ and the meaning of $\tau_{spar}=10$.**
>
> $\tau_{spar}$ is a pre-defined target sparsity in training. The dynamic controller adjusts the sparsity weight to keep the average $L_1$ activation near this target, so $\tau_{spar}=10$ means an average $L_1$ activation magnitude of approximately 10.
>
> >**Q4: Baseline setup and selection justification.**
>
> For the Token-SAE setup, please see our rebuttal in Q1 for reviewer 25Sd. We use Token-SAE as the main baseline because it is the standard token-level SAE and provides a controlled comparison for token vs. step granularity under the same backbone, data, and training setup. We do not include other SAE variants or logit-probing methods because they target different token-level objectives, such as feature discovery/steering (e.g., SAIF [1]) or decoding from dense hidden states (e.g., Patchscopes [2]), rather than explicitly capturing the semantics of an entire reasoning step.
>
> >**Q5: Unclear formulation of first-token perplexity prediction.**
>
> We do not predict perplexity directly. We predict the first-token distribution as a classification task, and compute perplexity from its cross-entropy on the ground-truth first token. Thus, “First Token Perplexity” reflects how well the latent feature supports next-step first-token prediction, rather than directly regressing perplexity.
>
> >**Q6: The statistical baseline is too naive and needs a stronger dense-state baseline.**
>
> In fact, your proposed baseline is already included in Table 1 as $h_k$. Specifically, $h_k$ is the final-layer dense hidden state and is probed with the same probing classifier as $\hat{h}_k$. The consistent gap between $h_k$ and $\hat{h}_k$ shows that although dense states contain relevant information, this information is more entangled and less directly decodable. In contrast, sparsity makes the reasoning signals more accessible and interpretable.
>
> > **Q7: Description of N2G.**
>
> Sorry for the confusion. N2G is a standard explainability tool that summarizes a latent’s activation behavior as a compact linguistic graph-like pattern. Due to space limit, we will include a definition of N2G in the discussion phase.
>
> > **Q8: Definition of ground-truth latent firings and alignment target.**
>
> We define ground-truth latent firings by thresholding each latent dimension’s activation value. We then compare these binary activations with the corresponding N2G pattern predictions using precision, recall, and F1.
>
> > **Q9: Claims of Table 4.**
>
> Table 4 is not meant to support the full claim by itself. Rather, the claim is supported jointly by Table 1, where sparse features better predict step-level attributes, N2G, where many features align with linguistic or reasoning patterns, and Table 4, which shows that SSAE features provide useful signals for inference-time weighting. We agree that other step-scoring methods can outperform ours in reranking; for example, our additional experiment in W1.2 for Reviewer MJTz shows stronger overall performance from a reward model. However, better reranking does not by itself imply a better step-level interpretable representation, since reward models are optimized for ranking quality rather than sparse feature disentanglement.
>
> > **Q10: Poor performance of Token-SAE in Table 1.**
>
> Thanks for the question! The naive baseline benefits from **dataset-level** priors, while Token-SAE is still **token-level**, so step-level information may remain entangled or only weakly expressed in the selected token feature, which explains why it may sometimes match or underperform the naive baseline. We agree that stronger methods, such as stronger probes, could do better than both. Our goal here **is not** to claim Token-SAE is weak, but to highlight that step-level sparse features make reasoning-related properties much more directly accessible than token-level features under the same probing setup.
>
> [1] SAIF: A sparse autoencoder framework for interpreting and steering instruction following of language models.2025
>
> [2] Patchscopes: A unifying framework for inspecting hidden representations of language models.2024

---

> > ### Author Rebuttal · Reviewer_SZjo · 2026-04-01
> >
> > While my concerns regarding baselines and related work have mostly gone un-addressed, the rest have. I think the empirical results, with the clarifications given by the authors, are convincing enough.

---

> > > ### Author Response · Authors · 2026-04-02
> > >
> > > Thank you for your thoughtful follow-up and updated assessment! We are glad that our clarifications were helpful, and we appreciate your constructive feedback.

---

### Official Review · Reviewer_MJTz · 2026-03-05

**Soundness:** 2
**Presentation:** 2
**Significance:** 3
**Originality:** 3
**Overall Recommendation:** 3
**Confidence:** 3

**Summary:**

This paper focuses on the interpretability of large language models, with a particular emphasis on the issue of granularity mismatch present in existing SAE methods. In contrast to traditional SAEs that work at the token level, the authors contend that reasoning dynamics are more effectively modeled at the step level. To resolve this granularity mismatch, they introduce the SSAE. The SSAE framework incorporates a context-conditioned encoder and decoder, which leverage prior context to extract the incremental information of each reasoning step into a sparse feature vector. The effectiveness of the SSAE is validated via diagnostic probing, Neuron-to-Graph-based pattern mining, and feature manipulation. Furthermore, the authors present a Probe-Guided Weighted Voting approach, which enhances inference performance on reasoning benchmarks including GSM8K and MATH-500.

**Compliance With Llm Reviewing Policy:**

Affirmed.

**Key Questions For Authors:**

See the Weaknesses.

**Limitations:**

Yes

**Strengths And Weaknesses:**

Strengths：

1. The shift from token-level to step-level analysis is well-motivated. Reasoning in CoT is inherently structured in steps, and analyzing individual tokens often results in polysemantic noise that is hard to interpret. Addressing the "incremental information" specifically is a sound theoretical approach.

2. The paper covers a wide range of evaluation methods, from quantitative probing to qualitative pattern mining and downstream application. The use of N2G to visualize the semantic meaning of the learned sparse features adds depth to the interpretability claims.

3. The proposed Probe-Guided Weighted Voting demonstrates that the learned features have practical utility beyond just visualization. Improving over the Self-Consistency baseline on benchmarks like GSM8K and AIME indicates that the SSAE features capture signals relevant to the validity of the reasoning process.

Weaknesses：

1. The comparison with "Token-SAE" in Table 1 seems potentially unfair or at least ill-defined. A single token's SAE feature obviously cannot capture the semantics of a whole sentence/step. A more appropriate baseline would be an aggregation of Token-SAE features over the step, or simply standard sentence embeddings. Furthermore, regarding the downstream task of "Probe-Guided Weighted Voting", the paper compares against Self-Consistency. However, the standard approach for weighting or reranking reasoning paths is using a Reward Model or a verifier trained on the dense hidden states.

2. While the application to "steps" is new, the architecture itself is a standard practice in conditional generation. Essentially, the method treats the hidden state of the last token of a step as the input to an SAE. The "context-conditioned" aspect is necessary for the decoder to function autoregressively, but it doesn't represent a significant algorithmic innovation in the realm of dictionary learning.

3. The definition of a "step" is highly dependent on the dataset. The paper relies on datasets where steps are clearly demarcated. It is unclear how SSAE generalizes to more fluid, unstructured reasoning or general text generation where "steps" are not explicitly defined.

---

> ### Author Rebuttal · Authors · 2026-03-31
>
> We are deeply grateful for your thorough and insightful feedback. Your expertise and dedicated time have significantly contributed to improving the quality of this work!
>
> > **W1.1: Is the Token-SAE baseline a fair comparison?**
>
> Thanks for your insightful question! When designing the experiment, we took into account the same concerns as yours. To make the comparison fair, we extract the Token-SAE feature from the **very last token of each reasoning step**, which has been widely used as a sentence-level representation (i.e., sentence embedding as you mentioned) in autoregressive models. We then fed this last-token feature into the same 3-layer MLP probes used for our SSAE features to predict the step-level attributes (as reported in Table 1).
>
> > **W1.2: Comparing Probe-Guided Weighted Voting with reward models or verifiers.**
>
> Thanks for your suggestion! We fully agree that Reward Models or verifiers trained on dense hidden states are the standard paradigm for reasoning path reranking, and we therefore add a comparison between our “Probe-Guided Weighted Voting” method and a pretrained Reward Model (Math-Shepherd-Mistral-7B-PRM), as shown in the table below.
> |Model|Strategy|GSM8K|SVAMP|MultiArith|
> |-|-|-:|-:|-:|
> |Qwen2.5-0.5B|RM@16|54.80|39.33|61.11|
> |Qwen2.5-0.5B|PG@16 (ours) |46.80|33.00|61.67|
> |Llama3.2-1B|RM@16|22.00|51.00|90.56|
> |Llama3.2-1B|PG@16 (ours) |19.40|50.33|83.89|
>
> Our method achieves compatible performance on MultiArith with Qwen2.5-0.5B and SVAMP with Llama3.2-1B, while underperforming the reward model in the remaining settings. However, our goal in this work **is not** to achieve SOTA step evaluation accuracy, but rather to demonstrate that SSAE features already contain useful and accessible signals for this purpose. Meanwhile, compared with our SSAE, RMs often require deploying a separate LLM of comparable size to the generator, or at least running a full heavy forward pass with a complex value head, which significantly increases memory overhead and inference latency. In contrast, our approach only needs a lightweight 3-layer MLP probe, whose runtime is negligible and has high computational and memory efficiency.  Meanwhile, traditional RMs are difficult and expensive to train because they require massive amounts of fine-grained, step-level human annotations, but our method is trained purely on a self-supervised reconstruction objective. The fact that a simple, lightweight probe can accurately predict step correctness from $\hat{h}_k$ (as shown in Table 1) demonstrates a profound interpretability finding: the LLM already internally encodes the correctness and logicality of its reasoning steps.
>
> > **W2: Algorithmic innovation vs conditional generation.**
>
> Thanks for your comment! We agree that conditional generation itself is not novel; however, our contribution lies in constructing a structured information bottleneck in conditional generation that **precisely extracts the incremental information of each reasoning step**, which is particularly useful for analyzing step-level semantics in LLMs. Specifically, the context-conditioned decoder in SSAE is not merely for autoregressive reconstruction. Its key role is to offload background information to the context, thereby forcing the sparse latent vector to encode only the incremental information of the current reasoning step. This creates an explicit information decomposition, which is fundamentally different from standard SAE setups that entangle both.
>
> As a result, SSAE learns step-level, semantically meaningful features, which cannot be captured by token-level SAEs, as shown in Table 1. Therefore, the novelty is not in conditional generation, but in using it to enable incremental, step-aligned representation learning for reasoning processes, which is not addressed in prior works.
>
> > **W3: Definition of "step" in other datasets.**
>
> Thanks for your question! We would like to clarify that in our framework, a “step” is not tied to any dataset-specific annotation, but is treated as a sentence-level unit. In practice, datasets such as NuminaMath-CoT and GSM8K do not explicitly define “steps”. We segment the text into steps based on natural linguistic boundaries (e.g., punctuation or line breaks). From this perspective, SSAE does not rely on specially structured data. Instead, it operates on a general formulation where each unit $s_k$ is a segment of text (e.g., a sentence), and the model learns to capture the incremental semantic update from the preceding context. This formulation is also applicable to more fluid or unstructured text, as long as the data can be segmented into contiguous textual units.
>
> Therefore, SSAE is inherently compatible with general text generation settings, as it only requires a context–segment decomposition, which can be obtained through standard sentence segmentation. More advanced or task-specific segmentation strategies could be incorporated in future work, but do not require any change to the core method.

---

### Official Review · Reviewer_25Sd · 2026-03-11

**Soundness:** 2
**Presentation:** 2
**Significance:** 2
**Originality:** 3
**Overall Recommendation:** 4
**Confidence:** 3

**Summary:**

This work proposes Step-Level Sparse Autoencoder (SSAE) as an improvement to the token-level sparse autoencoder (Token-SAE). The key differences are: 1. SSAE reconstructs a reasoning step instead of a single token. 2. Both the encoder and the decoder of SSAE take the past reasoning steps as additional input. Experiments demonstrate that the features extracted by SSAE can better predict some step-related features of the reasoning, including the correctness of the step. Using this information, the author proposes to do a weighted majority voting with the predicted correctness score as the weight signal, which achieves better performance comparing to a naive majority voting.

**Compliance With Llm Reviewing Policy:**

Affirmed.

**Final Justification:**

The paper proposes a moderately novel method to improve Token-SAE. I was concerned about the weak baseline and insufficient ablation study, but in the rebuttal, they address these concerns by providing a stronger baseline and more ablation study results, which improves the soundness of the paper. They also clarified that the weak results on larger models are not trained on those larger models, so the they are not evidence of bad scalability. Overall, I think the paper can get a score of 4.

**Key Questions For Authors:**

1. How do you use Token-SAE to predict step-level features? For example, are you averaging over presentations of all the tokens in one step?
2. Have you tried a larger tau? Why does sparsity matter in this work?
3. Is there a specific reason that you always set c=1? Does a larger c impact the performance?

**Limitations:**

yes

**Strengths And Weaknesses:**

Strengths:
1. The author did a lot of explorations based on this idea. The experiment section is rich and informative.
2. The method seems reasonable and moderately novel.

Weaknesses:
1. The presentation is not very clear. For example, it's not clear how the authors use Token-SAE to predict step-level features. The soundness is therefore impacted as that is the most important baseline.
2. Some discussions are not deep enough. For example, in section 4.3, the authors find that tau=10 performs better than 5 and 3, then jumps to the conclusion that a higher dimensionality is better --- then why not use a larger tau? Why do we need this sparsity constraint anyway?
3. The result in the appendix seems not very strong, indicating that the method might only work with smaller model.

---

> ### Author Rebuttal · Authors · 2026-03-31
>
> We are deeply grateful for your insightful and thorough feedback, and we appreciate the recognition of our work's contribution! The suggestions and comments made for our work have significantly helped to improve its quality.
> > **Q1 & W1: Implementation of Token-SAE.**
>
> Thank you for the suggestion. We trained a Token-SAE using Qwen2.5-0.5B as the backbone. Since Token-SAE produces a sparse latent for each token, there is an inherent granularity mismatch for predicting step-level attributes. To construct a fair baseline, we use the Token-SAE sparse feature of the **last token of each reasoning step**. This last-token feature is then fed into the same 3-layer MLP probes used for SSAE to predict the step-level attributes reported in Table 1.
>
> > **Q2 & W2: Why do we need the sparsity constraint anyway? Why not use a larger tau?**
>
> Thank you for your insightful questions!
>
> The sparsity constraint is essential for two reasons. First, it prevents the latent from degenerating into a dense identity-mapping representation. Instead, it extracts only the incremental information of the current reasoning step, filtering out the background context. Second, sparsity is what makes the features more disentangled and interpretable, which is the core motivation of both SSAE and SAEs more broadly. In our framework, this is implemented through the $L_1$ penalty and the information bottleneck in Sec. 3.1 / Eq. 8.
>
> Regarding larger $\tau_{spar}$, it controls the average number of active dimensions per step and thus the trade-off between capacity and interpretability. A larger $\tau_{spar}$ widens the information bottleneck. This may allow more background noise to leak back into $\hat{h}_k$ and spread one step’s information across multiple dimensions, making each dimension less coherent and reducing interpretability.
>
> We further evaluated larger $\tau_{spar}$ values with N2G on SSAE-Qwen, G/N denote GSM8K/NuminaMath-CoT, and P/R/F1 denote Precision/Recall/F1.
> | $\tau_{spar}$ | G-P | G-R | G-F1 | N-P | N-R | N-F1 |
> |---|---:|---:|---:|---:|---:|---:|
> | 25 | **90.14** | 64.03 | 74.87 | **96.53** | 87.29 | 91.68 |
> | 15 | 80.58 | 74.44 | 77.40 | 90.99 | **95.18** | **93.03** |
> | 10 | 79.00 | **80.00** | **79.49** | 74.23 | 72.45 | 73.33 |
>
> On GSM8K, $\tau_{spar}=10$ gives the best overall balance: although $\tau_{spar}=25$ achieves higher precision, its recall and F1 drop, suggesting that the learned features become more selective but also more fragmented, often capturing narrower phrase-level or numeric patterns. As a result, the induced N2G graph becomes more conservative: its predictions are more often correct, but cover a smaller fraction of the true activations. This hurts overall step-level interpretability. On NuminaMath-CoT, a moderate increase to $\tau_{spar}=15$ is beneficial, likely because its reasoning steps are longer and more compositionally complex, so $\tau_{spar}=10$ is somewhat under-capacity. However, performance drops again at $\tau_{spar}=25$, indicating that excessively large $\tau_{spar}$ is not uniformly better; the optimal value depends on balancing representational capacity against coherent, interpretable step-level features.
>
> > **W3: The appendix results on larger models**
>
> Thank you for this comment. Our goal **is not** to achieve SOTA reranking performance on larger models, but to show that SSAE features already contain useful and accessible reasoning signals, and these signals can extend to larger models to some extent **without additional training**. The appendix serves as a cross-model transfer test: SSAE and probe are trained on small backbones, then used to score reasoning paths from much larger models.
>
> Under this setting, the appendix still shows some transferability, suggesting that the sparse step-level signals learned by SSAE are not confined to the training backbones. Moreover, without the cross-model transfer (Table 4), PG consistently improves over Self-Consistency on both Qwen2.5-0.5B and Llama-3.2-1B, showing that lightweight probes on SSAE features already extract reasoning-relevant information for inference-time weighting.
>
> > **Q3: Why set $c=1$? Does a larger $c$ impact the performance?**
>
> Thanks for your insightful question. We set $c=1$ to maintain a minimal and controlled latent capacity, so that SSAE focuses on encoding the most salient incremental information of each reasoning step rather than distributing it across multiple segments. Empirically, $c=1$ already yields strong probing and downstream performance, showing that the formulation is effective without requiring additional capacity.
>
> A larger $c$ would expand the latent capacity and relax the bottleneck. which may capture more fine-grained features but could also introduce redundancy and reduce interpretability. We agree that this is an important factor to study, and we will include an ablation on different values of $c$ in the revision to better understand its impact on both performance and feature disentanglement.

---

> > ### Author Rebuttal · Reviewer_25Sd · 2026-04-01
> >
> > Most of my concerns are addressed. But I feel that the comparison against Token-SAE seems not very fair. I mean, it's very natural that the information in a single token -- no matter whether it's the last token of a reasoning step -- is not enough to predict the information of the whole reasoning step. Some stronger baselines would be training a model to aggregate over the Token-SAE representations of tokens in a reasoning step > use sentence-embeddings > use some heuristics to aggregate over the Token-SAE representations of tokens in a reasoning step.

---

> > > ### Author Response · Authors · 2026-04-03
> > >
> > > Thank you very much for your constructive feedback and your recognition that most of your concerns have been addressed. For your remaining concern, following your suggestion, we add a baseline that trains a transformer aggregator to aggregate over the Token-SAE sparse representations of tokens in a reasoning step.
> > >
> > > Specifically, we keep the same “Qwen2.5-0.5B + Token-SAE” backbone, but instead of using only the last-token sparse feature, we feed the **full** sequence of Token-SAE sparse features within a step into a transformer aggregator. The aggregator uses positional embeddings and a 4-layer, 8-head Transformer, then mean-pools the outputs to produce a step-level vector, which is fed back to predict the step-level features in the same way as all other methods. Thus, this baseline provides a learnable aggregation module for Token-SAE, which we believe can directly address your concern.
> > >
> > > In the table, Corr denotes correctness accuracy, Logi denotes logicality accuracy, StepErr denotes step-length error, and PPL denotes first-token perplexity. G / M refers to GSM8K / MATH-500, respectively.
> > >
> > > | Model | Corr-G | Corr-M | Logi-M | StepErr-G | StepErr-M | PPL-G | PPL-M |
> > > |:--|--:|--:|--:|--:|--:|--:|--:|
> > > | Token-SAE-last | 72.44 | 86.79 | 60.56 | 29.06 | 31.58 | 103.54 | 49.17 |
> > > | Token-SAE-agg | 74.38 | **86.88** | 67.43 | 25.79 | 30.33 | 61.55 | 18.63 |
> > > | SSAE-Qwen | **78.58** | 82.74 | **76.56** | **2.10** | **1.94** | **4.09** | **1.46** |
> > >
> > > From the table above, this baseline indeed improves substantially over the original Token-SAE-last-embedding baseline. However, even with this stronger aggregation, SSAE still performs better overall on the most important step-level attributes, especially logicality and the two structural prediction tasks. This suggests that SSAE’s advantage comes not merely from aggregation, but from learning sparse features directly at the step level. By contrast, Token-SAE is fundamentally trained on token-level activations and thus remains mismatched to step-level prediction even when aggregating all token-level features in the sentence.
> > >
> > > According to your suggestion, we will add this stronger Token-SAE aggregation baseline and corresponding discussion in the revised paper.

---

### Decision · Program_Chairs · 2026-04-30

**Decision:**

Accept (regular)

**Comment:**

This paper proposes the Step-Level Sparse Autoencoder (SSAE), a context-conditioned SAE variant that operates on reasoning steps rather than individual tokens. The context-conditioned encoder and decoder enable the sparse latent to encode only the incremental information in each step. The paper validates SSAE through probing experiments (Table 1), Neuron-to-Graph pattern mining, and a Probe-Guided Weighted Voting inference strategy (Table 4).

**Strengths.** Reviewers agree that the shift from token-level to step-level analysis is well motivated, since reasoning in CoT is structured in coherent steps rather than individual tokens (MJTz, SZjo). The evaluation covers probing, pattern mining, and a downstream Probe-Guided Weighted Voting strategy. The rebuttal significantly improved the paper: it clarified architectural details (which led goJi to raise their score to 5), addressed baseline concerns by providing a new Token-SAE-agg baseline (which convinced 25Sd to raise their score to 4), and added a Reward Model comparison for the voting experiment.

**Weaknesses.** Some limitations remain regarding empirical scope and generalizability. First, despite the stronger baselines added during rebuttal, Reviewer SZjo maintains that the overall baseline breadth for the main experiments could be further expanded to fully substantiate the paper's claims. Second, the Probe-Guided Voting gains over Self-Consistency are somewhat marginal ($+0.6$ to $+3.3$ points) and the newly added Reward Model baseline outperforms SSAE in several settings, although reviewers acknowledge SSAE's efficiency advantages. Finally, Reviewer MJTz notes that the current definition of a "step" is dataset-dependent, and it remains unclear how SSAE generalizes to unstructured reasoning where step boundaries are not explicitly demarcated.

**Decision.** I recommend **accept (weak accept)**. Three of the four reviewers support acceptance with post-rebuttal scores of 4, 5, and 5. Reviewer MJTz remained at 3 and did not engage with the rebuttal, but the authors did substantively complete the stronger baseline experiments MJTz requested. The conceptual motivation is strong and the rebuttal successfully addressed the most critical soundness concerns. For the camera-ready version, the authors are expected to incorporate: (i) the Token-SAE-agg baseline results and the Reward Model comparison provided during the rebuttal into the paper; (ii) a discussion of how the step definition might generalize beyond datasets with explicit delimiters; and (iii) a PG@$N$ vs SC@$N$ scaling curve at varying $N$ (e.g., $N \in \{4, 16, 64, 128\}$).